# Artificial Intelligence in Head and Neck Cancer: Towards Precision Medicine

**DOI:** 10.3390/cancers17183023

**Published:** 2025-09-16

**Authors:** Jacob Hagen, Logan Hornung, William Barham, Supratik Mukhopadhyay, Adam Bess, Kevin Contrera, Devraj Basu, Vlad Sandulache, Guillaume Spielmann, Sagar Kansara

**Affiliations:** 1Department of Otolaryngology-Head and Neck Surgery, Louisiana State University Health Sciences Center, New Orleans, LA 70112, USA; jhage2@lsuhsc.edu (J.H.); lhornu@lsuhsc.edu (L.H.); wbarh1@lsuhsc.edu (W.B.); 2Department of Environmental Sciences, Center for Computation & Technology, Louisiana State University, Baton Rouge, LA 70803, USA; mmukho1@lsu.edu (S.M.); adam.bess@gmail.com (A.B.); 3Department of Otolaryngology-Head and Neck Surgery, University of Pittsburgh, Pittsburgh, PA 15260, USA; contrerakj@upmc.edu; 4Department of Otolaryngology-Head and Neck Surgery, University of Pennsylvania, Philadelphia, PA 19104, USA; devbasu@pennmedicine.upenn.edu; 5Department of Otolaryngology-Head and Neck Surgery, Baylor College of Medicine, Houston, TX 77030, USA; vlad.sandulache@bcm.edu; 6School of Kinesiology, Louisiana State University, Baton Rouge, LA 70803, USA; gspielmann@lsu.edu

**Keywords:** artificial intelligence, machine learning, deep learning, artificial neural network, head and neck cancer, squamous cell carcinoma

## Abstract

This review was conducted to highlight current and potential applications of artificial intelligence (AI) in the care of patients with head and neck cancer, the barriers to its clinical implementation, and the relative scarcity of research in this field. While AI has demonstrated success in other areas of oncology, its role in head and neck cancer remains limited, with progress slowed by technical, ethical, and logistical challenges. Nevertheless, integrating AI with multiomics approaches presents a promising opportunity to advance precision medicine in this patient population. The findings presented in this manuscript may encourage further multi-institutional studies aimed at validating AI-driven strategies and accelerating their adoption in clinical practice.

## 1. Introduction

### 1.1. Epidemiology and Economic Burden

Head and neck cancer (HNC) encompasses malignant neoplasms originating from the soft tissues of the nasal cavity, paranasal sinuses, oral cavity, pharynx, larynx, skin, and thyroid. According to the National Cancer Institute’s Surveillance, Epidemiology, and End Results (SEER) program, cancers arising from these sites have a current prevalence of over 1.5 million and a total incidence rate of 28 per 100,000 as of 2021 [1]. Since 2000, the incidence rate for HNC has increased 23.9% [1]. The median age at diagnosis is 64 to 66 years for most mucosal cancers, 51 years for thyroid cancer, and 60–66 years for most skin cancers. HNC has been shown to be 3–4 times more common in men [1].

Historically, tobacco and alcohol use have been the primary risk factors for HNC, with a well-documented synergistic effect on disease risk. However, in developed countries, despite declining tobacco use, the incidence of oropharyngeal and thyroid cancer continues to rise. This shift has led to the recognition of human papillomavirus (HPV) as a significant risk factor in head and neck squamous cell carcinoma (HNSCC). Notably, the incidence of HPV-associated oropharyngeal HNSCC now surpasses that of tobacco- and alcohol-related HNC [2]. HPV associated HNSCC has reached epidemic proportions [3].

SEER data suggest that the stage at diagnosis of HNC varies by site and significantly influences survival rates. Laryngeal cancer is nearly twice as likely to be detected at a localized stage than oral cavity and pharyngeal cancers (29.5% vs. 50.6%). Survival outcomes tend to correlate with the stage at diagnosis as shown by the more favorable survival for cancers that diagnosed earlier in the disease course, with five-year survival rates ranging from 74.0 to 87.5% for localized mucosal cancer stage vs. 32.6 to 37.5% for distant cancer [1]. However, despite advances in our understanding of HNSCC, oncologic outcomes have remained largely unchanged for the past several decades.

### 1.2. Clinical Challenges in HNC and the Role of AI

The complexity of HNC creates significant roadblocks in the diagnosis, treatment, and surveillance of patients. Intervening in a timely manner is crucial, but delays are often found throughout the care continuum. Prior to diagnosis, the earliest symptoms develop gradually and are often subtle. Unlike breast or colon cancer, there are no standardized screening guidelines for HNC. This often leads to many diagnoses occurring at later stages. The diagnostic phase is further hindered by limited accessibility to several key procedures including pathology processing, computed tomography (CT), magnetic resonance imaging (MRI), positron emission tomography (PET), and tumor biomarker testing. Treatment planning is similarly complex as it frequently requires a combination of interventions such as surgery, chemoradiation, and immunotherapy, which all require a high degree of inter-specialty collaboration and monitoring for adverse effects. Personalized treatment is essential, as there are many variables such as staging and comorbid conditions which are unique to each patient and can impact the disease course. Following treatment, further hurdles arise in care coordination, recurrence monitoring, and managing electronic health records (EHRs), all of which may slow recognition and treatment of high-risk individuals.

AI continues to emerge as a powerful tool in medicine and has exhibited significant utility in all phases of cancer treatment including screening, diagnosis, treatment, and surveillance. Early investigations into the application of AI in management of HNC have suggested that it can improve accessibility across the care continuum. These abilities reduce the time to intervention and enhance personalized treatments, which may potentially lead to better patient outcomes. This review outlines AI’s role in HNC, beginning with an overview of several AI learning principles and their utility in the field, followed by an examination of common challenges seen with AI’s integration in HNC and areas for future development. Figure 1 below outlines the HNC care continuum and areas where AI can be applied.

## 2. An Introduction to AI in Medicine

AI is a multidisciplinary field that uses statistical approaches to analyze large datasets and identify patterns in ways that resemble the human decision-making process [4]. Several methods of AI learning contribute to this goal, including machine learning (ML) and deep learning (DL), which involve artificial neural networks (ANNs). An AI algorithm can take various modalities as inputs, such as text, video, images, graphs, and tables. Each method possesses its own data inputs, operational parameters, and objectives [5].

ML relies on three core components: a dataset, an optimization algorithm, and a target output. The primary objective is to build a model that can ascertain the existing mapping between the input and target output, which is accomplished by minimizing the distance between the ML’s predictions and the ground truth [5]. The mapping function obtained can then be applied to unseen datasets to infer patterns and make predictions. Within ML, there are four key techniques by which this process occurs: supervised, unsupervised, semi-supervised, and reinforcement learning each defined by how the input are structured and the manner in which the model learns from them [4,5]. An additional method exists, called Foundation models, which incorporate elements of each of the four listed key techniques into a singular model [6].

Supervised learning involves using labeled datasets to train models that identify associations between input data and their corresponding target labels. Once these associations are learned, the model can be applied to new, unseen data not included in the original training set and without predefined labels to determine whether it fits the learned mapping function. In contrast, unsupervised learning analyzes unlabeled data using algorithms to identify patterns and group data based on discovered relationships, a process known as “clustering.” Semi-supervised learning applies when a dataset contains both labeled and unlabeled data. The model is initially trained on the labeled subset and then uses the learned associations to infer labels for the remaining unlabeled subset. Because these assignments are inferred rather than confirmed, they may not always align with the ground truth [4,5]. Reinforcement learning differs from the above approaches by training a model to achieve a desired outcome without predefined parameters. The algorithm is learned through a trial-and-error process, receiving feedback in the form of “rewards” or “penalties.” This technique resembles certain aspects of human cognition, as it involves learning through interactions with an environment that provides positive or negative reinforcement, guiding the model in a manner that increases the likelihood of achieving the desired outcome [4,5].

DL, a subset of ML, is a powerful technique used for data analysis, pattern recognition, and classification. It operates through ANNs, which are loosely modeled after the human neural networks. At its core, an ANN consists of layers of interconnected nodes, or “neurons”, that transfer information between themselves in a unidirectional or bidirectional manner. The input layer receives raw data, multiple hidden layers perform progressively complex computations, and the output layer synthesizes the processed information to classify it accordingly. If the algorithm arrives at an incorrect classification, user feedback triggers an adjustment to the weight, or significance, of each input the node uses when relaying information. This process of feedback and adjustment allows the model to progressively improve its accuracy. Deep neural networks, a specific type of ANN, are widely used for image identification, classification, and pattern recognition [4,5].

Natural language processing (NLP) is one of the most well-studied areas of ML. As the name suggests, NLP focuses on enabling machines to receive, process, understand, and generate human language in an effective way. This is accomplished through the analysis of large volumes of linguistic input, typically in the form of text or spoken language. Its utility is broadly applicable and best observed in tasks such as text summarization, generation, and refinement [7,8,9].

Foundation models are large-scale AI systems trained on massive, diverse datasets that can be adapted (fine-tuned) to a wide range of downstream tasks. Their versatility comes from learning generalizable patterns across data, making them powerful tools for multimodal and domain-specific applications [6]. Table 1 below contains a brief summary of the common types of AI and areas of application. Within each modality of ML lie many different types algorithms designed for unique tasks. The most commonly encountered algorithms employed in HNC can be seen below in Table 2.

While AI demonstrates utility in many aspects of medicine, its integration into practice does not come without challenges or concerns. These challenges include data quality, algorithmic bias, transparency, security, and financial considerations.

## 3. Utility of AI in Imaging and Diagnostics

### 3.1. Artificial Intelligence in Radiology

CT, MRI and PET are three imaging modalities crucial to the management of HNSCC. AI has shown immense promise in several methods of medical imaging, particularly through deep learning techniques like convolutional neural networks (CNNs), which enhance diagnostic accuracy and efficiency. In HNC, imaging interpretation is often complicated by complex anatomy, subtle differences between benign and malignant tissue, and inter-operator variability, making AI a valuable tool to employ [13].

The use of ML models has significantly enhanced CT imaging in HNC, a key application of AI in terms of diagnostic capabilities, staging, and image enhancement. Zhao et al. trained convoluted neural networks (CNNs) on over 900 scans for thyroid nodule evaluation, with most models outperforming radiologists in sensitivity, specificity, and accuracy. Notably, CNN-generated heat maps highlighted both intra- and extra-nodular features, providing information beyond traditional reads [14]. Romeo et al. used texture analysis on CT images of oral cavity and oropharyngeal cancers, achieving over 90% accuracy in predicting tumor grade and nodal status [15]. Kann et al. similarly demonstrated high accuracy in detecting extra-nodal extension and lymph node metastases [16]. As ensuring accuracy and reducing inter-operator variability is crucial for diagnosing HNSCC, AI can enhance CT imaging through noise reduction, improved image quality, and more precise tumor segmentation [13]. The ability of AI to precisely segment tumors is especially important as it has implications beyond simply diagnosis.

MRI is a valuable tool in HNC imaging but has several limitations. One major drawback is the prolonged acquisition time, which is necessary to capture the fine anatomical details of the head and neck. As the anatomical region under observation decreases in size, the degree of movement, and subsequent artifact produced is magnified. When considering this in addition to the time required to obtain imaging, it becomes apparent that reducing scan time and motion-related artifacts is important in improving image quality. To address this, compressed sensing techniques have emerged over the past 15 years to reduce scan times. Recently, DL-based denoising methods have been used alongside compressed sensing to improve image quality and shorten scan time [17]. Naganawa et al. demonstrated this potential in a retrospective study of endolymphatic hydrops MRI scans, where the use of AI-based denoising led to a fourfold increase in contrast-to-noise ratio [18]. Building on this, a 2025 study by Fujima et al. applied deep learning reconstruction and super-resolution techniques to fat-suppressed T2-weighted head and neck MRIs with resulting improved image quality, higher signal-and contrast-to-noise ratios, and reduced acquisition times compared to traditional methods [19].

PET plays a key role in diagnosis, staging, radiation planning, response assessment, and recurrence monitoring of HNC. PET imaging enhanced by AI and radiomics has been studied for its ability to distinguish between benign and malignant tissue [20], predict treatment response [20], progression-free survival [21], and detect lymph node metastasis [16,22]. DL models have demonstrated diagnostic accuracies of 90–100% in distinguishing malignancies such as oropharyngeal HNSCC, thyroid follicular carcinoma, and mucoepidermoid carcinoma from benign tissue [20], highlighting the potential of AI-enhanced PET in evaluating both primary tumors and nodal disease.

AI-based radiomic algorithms trained on PET scans of patients with hypopharyngeal and laryngeal cancers treated with varying modalities predicted one-year treatment response with 80% to 90% accuracy [23]. Of note, in oropharyngeal HNSCC, AI models have predicted progression-free and overall survival with greater consistency than the American Joint Committee on Cancer (AJCC) staging system [21].

While these studies demonstrate encouraging results, the current body of research on AI applications in PET and other radiologic imaging modalities specifically within HNC remains limited, with most studies involving relatively small patient cohorts and narrow clinical focus. Expanding research efforts in this area may enhance diagnostic, prognostic, and treatment-planning capabilities, as well as generalizability.

### 3.2. Artificial Intelligence in Histopathology and Biomarker Assessment

Given the ability of AI to detect complex patterns and associations within images, there has been increased interest in training and validating DL models to be used in histopathology [24,25]. These applications include cancer detection and grading, prognostic prediction, and quality assurance. The integration of AI in pathology has provided great benefits by improving efficiency and accessibility.

Recent studies have demonstrated high diagnostic accuracy of AI models across many types of cancer. Top-performing algorithms achieve accuracies of up to 99% [26]. For instance, Guan et al. used deep convolutional neural network (DCNN) on H&E-stained thyroid nodule FNA specimens from patients with and without papillary thyroid carcinoma (PTC). Their highest-performing model diagnosed PTC with 97.6% accuracy, 100% sensitivity, and a negative predictive value of 100% [27]. Feature analysis showed the algorithm relied on nuclear contouring, crowding, and perimeter, which are features commonly noted by radiologists, as key indicators. Notably, the DCNN also performed pixel-level analysis of staining intensity and detected consistently higher intensity in PTC compared to benign nodules [27]. While these histological features are not new, the ability of AI to analyze slides on a granular scale surpasses human visual capabilities, allowing the detection of subtle variations that may otherwise go unnoticed.

While Guan et al. demonstrated the binary classification potential of AI (PTC vs. non-PTC), other studies have expanded research into tumor subtyping [27]. Wang et al., for example, used DCNN to differentiate among a spectrum of thyroid pathologies, including normal tissue, toxic adenoma, and the four major thyroid carcinomas—papillary, follicular, medullary, and anaplastic [28]. The model achieved impressive diagnostic accuracies ranging from 97 to 100% for carcinomas and normal tissue, and 92.4% for adenomatous lesions [28]. These findings highlight the strong diagnostic capabilities of AI, laying the foundation for its other applications in prognostic prediction and quality assurance.

AI in the form of CNN’s and supervised ML in conjunction with feature extraction have also been applied to histopathology of oral cavity SCC (OCSCC). For example, Fati et al. used a two-stage approach to investigate the ability of AI to analyze and diagnose OCSCC by histopathology. In the initial stage, a hybrid CNN/SML algorithm was trained, achieving diagnostic accuracy, specificity, and sensitivity of 98.1%, 98.4%, and 98.6%, respectively [29]. The second stage utilized transfer learning, in which features extracted from the hybrid algorithm were passed through an ANN tasked with identifying OCSCC; this stage achieved even higher accuracy, specificity, and sensitivity of 99.1%, 99.6%, and 99.5%, respectively [29].

Currently, tumor biomarker expression is assessed through peripheral blood tests or tissue sampling. However, recent studies have explored the use of AI in non-invasive biomarker expression, potentially enhancing prognostic assessment by integrating both pathomics and genomics data. For instance, Wang Y. et al. utilized a supervised ML algorithm, logistic regression (LR), to predict CXCL8 expression in HNSCC based on AI-extracted histological features, which demonstrated predictive performance comparable to traditional methods of expression analysis [30]. Given that CXCL8 has been associated with worse outcomes, the ability to detect expression is a demonstration of how AI can serve as an adjunct to the assessment of prognosis. In a separate study, Wang X. et al. used an unsupervised ML approach to predict CDKN2A expression, a tumor suppressor gene responsible for the creation of regulators of cellular division, in HNSCC, achieving accuracy levels approaching those of standard techniques [31].

Additionally, Bryan et al. demonstrated the diagnostic utility of AI in non-invasive biomarker assessment in HPV-associated HNSCC by comparing HPV-DeepSeek, an HPV genome sequencing assay liquid biopsy which uses ML, to traditional methods of liquid biopsy such as digital droplet PCR (ddPCR) and HPV serology [32,33]. In a dataset of over 150 samples, HPV-DeepSeek achieved a diagnostic sensitivity and specificity of 98.7%, which significantly outperforms ddPCR and HPV serology [32]. This study also highlighted the prognostic capabilities of AI as HPV-DeepSeek was able to detect high-risk genotypes and base-pair fragments which closely follow cancer staging [32].

The growing body of literature on AI in diagnostics has shown that AI models can perform with an accuracy and efficiency that exceeds traditional techniques. By streamlining digital pathology and radiology, AI has the potential to expand access to high-quality diagnostic technology. The consistency and speed of AI support its role in augmenting current diagnostic techniques as it also facilitates efficiency and quality assurance. Beyond individual applications within specific modalities, a multimodal approach that integrates genomics, radiomics, and pathomics with AI augmentation has the potential to greatly advance diagnostics and prognostication of HNC in ways previously thought unattainable.

## 4. AI in the Treatment Course

Radiation therapy is a cornerstone in the treatment of HNC. Providing radiation involves preparation, delivery, and continuous evaluation. Over the course of a patients treatment, anatomic changes often occur in both the tumor and the surrounding normal tissue. It is important to identify and adjust for even the most subtle of these changes in order to promote the most targeted and effective therapy possible. This concept is referred to as adaptive radiation. Adaptive radiation depends heavily on imaging, with CT used for dose recalculations and tissue contouring. Adaptive radiation is necessary for treatment optimization but often requires hours to generate and revise the contours of at-risk organs, followed by adjustments to radiation dosing. Additionally, this process is not exempt from demonstrating inter-operator variability. In recent years, there has been growing interest in the role of AI in enhancing adaptive radiation. In particular, AI-automated adaptive contouring has demonstrated potential to streamline this process and improve consistency [34]. For example, a 2023 study examined automated segmentation in HNC radiation planning and found that deep learning algorithms reduced the time needed to generate and revise organ-at-risk contours by 76% when compared to traditional methods [35]. Figure 2 below provides a visual representation of manual and AI-assisted adaptive radiation workflows.

While AI has demonstrated utility in many fields of medicine, its potential in surgery is becoming increasingly apparent. These applications center on improving precision, accuracy, and efficiency. One key area is surgical planning, which becomes exceedingly important when considering the complexity of the head and neck. AI’s ability to quickly analyze and segment structures with such high resolution and precision allows for a streamlined preoperative planning process. Furthermore, the speed and quality of image production and reconstruction may ease what is often a tedious process [36]. Beyond imaging, AI’s histopathological capabilities are being optimized for rapid intraoperative margin assessment [37]. AI-assisted pathology analysis has the potential to reduce intraoperative time, decrease the incidence of inadequate margins, and minimize the duration of anesthesia. Notably, several studies have investigated the use of ML in combination with hyperspectral imaging (HSI) to evaluate margins in HNC from resected specimens. These AI-assisted HSI systems have shown significant promise, with sensitivity and specificity nearing 90%, and rapid evaluation times measured in minutes [37,38,39]. With technologic advances in fields such as augmented reality, robotic assisted surgery (RAS) and intraoperative navigation modalities, the potential uses of AI in the operating room continue to expand [40]. For example, AI assistance in operative planning for image guided RAS has been shown to improve three-dimensional precision intraoperatively on three-dimensional mandibular tumor models [41]. AI has also been employed using endoscopic video footage for surgical skill assessment and has growing potential to serve as a tool for the improvement of surgical skills [42].

### Drug Discovery

Throughout the history of medicine, drug discovery has consistently been a catalyst for advancements. While much has changed regarding the process of drug discovery, its core components remain the same. Fundamentally, this process entails the identification of a target, finding a compound that can interact with the target, and determining if this interaction occurs in a desirable manner. As it stands, several gene expression databases, including the National Center for Biotechnology Information’s Gene Expression Omnibus (NCBI GEO) [43] and The Cancer Genome Atlas (TGCA) [44], and several genome wide association studies (GWAS) are routinely used in the identification of biomolecular targets [45]. Likewise, there are numerous accessible databases that make up the “virtual chemical space” [46]. While these banks of information are critical to the drug discovery process, they contain unfathomable amounts of information. The sheer volume of information to sift through represents a nearly insurmountable task, a task that artificial intelligence is well equipped for.

AI can enhance the efficiency and cost-effectiveness of drug discovery by advancing candidates with improved clinical tolerance through the development pipeline. For example, the DeepDrug group developed an AI-based pipeline that can not only develop novel small molecules but can also repurpose existing FDA-approved drugs for treating new diseases [47,48,49,50,51,52]. DeepDrug’s AI system enables a streamlined drug discovery process by utilizing mathematical models of interactions among all human proteins. Gupta et al. cite several instances in which gene expression databases and GWASs were used in conjunction with AI in the molecular target identification process [46]. For example, deep neural networks and ML algorithms have been exercised on these databases to identify targets in both cancerous and non-cancerous illnesses such as soft tissue sarcoma and JIA [46]. Once a target has been identified, the hunt for a molecule that can interact with this target begins. Over the past 10–15 years, ML has been implemented during this endeavor. Paul et al. recently discussed the vast nature of the virtual chemical space (containing over 10^6^ molecules) and how recently DL, as opposed to ML, has been employed, and has its demonstrated superiority [47]. The superiority of DL has not only been observed in the treatment identification process but also in modeling the quantitative structure–activity relationship, predicting ligand and receptor interactions, and forecasting the absorption, distribution, metabolism, excretion, and toxicity profile of the proposed treatment [47].

## 5. AI in Prognosis and Outcome Prediction, Risk Assessment, Patient Monitoring and Follow-Up

HNC presents significant challenges in prognosis due to its biological heterogeneity and high recurrence rates. AI, particularly ML, demonstrates promise as an innovative tool that can assist in enhancing survival predictions, recurrence risk assessments, and patient monitoring by efficiently analyzing and synthesizing large datasets comprising multifactorial data. These advancements would allow for improved clinical decision making and have the potential to wholly impact patient outcomes.

ML models have been shown to enhance survival predictions in HNC by rapidly analyzing extensive datasets comprising clinical, demographic, histologic, molecular, and imaging data. These models can then be used to generate survival curves and estimate individuals’ survival risks by identifying complex patterns among variables such as tumor stage, genetic markers, and treatment history-patterns that otherwise would not have been as easily identified through human analysis alone [5,53]. For instance, ML models have demonstrated high accuracy in predicting recurrence risks and survival rates for laryngeal squamous cell carcinoma (LSCC) and oropharyngeal squamous cell carcinoma (OPSCC) by integrating and synthesizing multifactorial data in a manner that far surpasses traditional statistical approaches [53,54,55]. ML’s ability to process complex datasets and identify subtle relationships between prognostic factors, such as specific genetic mutations or histomorphometric features that are associated with more aggressive versus indolent tumors, further demonstrates its clinical and prognostic utility.

Pathomics, combined with AI, has demonstrated its utility for analyzing large datasets of histologic images and identifying histomorphometric features in premalignant lesions associated with risk for malignant transformation. In a study published by Cai et al., pathomics was used to stratify patients with premalignant oral leukoplakia into two groups, high versus low risk for malignant transformation, and then predict the likelihood of malignant transformation. The model used in this study outperformed a group of experienced pathologists with an AUC of 0.84 as opposed to 0.73 by experienced pathologists. Additionally, the study attempted to address the “black box”-poor interpretability of models generated by deep learning and AI-by using IHC staining of biomarkers (Ki67, p53, and PD-L1) implicated in epithelial dysplasia which showed correlations with the pathomics features [56]. This points to AI’s usefulness as a tool to enhance screening and surveillance of premalignant lesions.

Radiomics, combined with AI, has showcased its utility for extracting quantitative features from imaging data that can be used to further refine survival prediction models [57]. In HNC, where tumor heterogeneity complicates prognosis, radiomics-based models provide strong outcome predictions, supporting the planning of personalized treatment. Continuous model updates using real-time data entry ensure that predictions remain accurate, optimizing clinical decision making and patient-counseling.

AI-driven models are a critical tool for assessing HNC recurrence risk and tackling the challenges posed by tumor heterogeneity and incomplete surgical resection. ML is trained on large datasets to develop algorithms that systematically analyze imaging, biomarkers, histological, and genetic data, identifying recurring patterns among these variables that are associated with HNC, which may enable early detection [5]. For example, AI has been used to detect subtle changes and irregularities, in datasets comprising advanced imaging studies-such as CT or MRI-histologic data, and biomarker profiles, that can facilitate more timely interventions [53,58].

AI has demonstrated its efficacy in post-treatment surveillance and recurrence monitoring as supported by several recent studies. Fatapour et al. demonstrate how a ML model for detecting oral squamous cell carcinoma of the tongue successfully integrated clinical and histopathological data with a high degree of sensitivity [59]. Similarly, López-Cortés et al. highlighted AI’s ability to accurately identify high-risk patients, reducing missed recurrences and optimizing follow-up strategies [60]. These advancements underscore AI’s potential role in optimizing HNC management and patient-centric care through early identification of high-risk patients.

AI has the potential to enhance HNC monitoring by integrating data from EHRs and wearable devices to track disease progression and treatment adherence [5]. Real-time analysis of mobile health application data and wearable sensor outputs may enable early detection of recurrence through monitoring changes in vital signs or biomarkers [61]. Additionally, AI-driven tools can be used to monitor medication compliance and deliver personalized interventions, such as smartphone-based reminders, to address non-adherence [62]. These tools can assist patients and clinicians in the pursuit of data-driven, proactive care. Specific state-of-the-art algorithms and their performances can be found below in Table 3.

## 6. AI in Precision Medicine

Catalyzed by the advent of genomic sequencing, precision medicine is a constantly expanding area of medicine aimed at providing highly specific diagnoses, tailored treatment plans, and accurate outcome predictions. This progress is made possible through modern genomic sequencing, biomarker analysis, and EHR [63,64]. However, sifting through a patient’s genome, vast biomarker data, and the overwhelming volume of EHR documentation is a time-intensive and cumbersome task. The recent advancements in AI learning techniques have introduced a valuable tool that is proving to be indispensable in making this process more efficient and expedient. AI’s ML, DL, and NLP capabilities enable the rapid analysis of vast datasets, thereby facilitating personalized medicine, especially as it relates to diagnostic specificity, treatment planning, and outcome prediction [63,64]. By integrating phenotypic documentation and medical imaging from EHRs with genomic data, AI can analyze several components in concert, optimizing the field of precision medicine to its fullest potential.

AI has the potential to be used in precision medicine through multiomics applications. Its use in genomic analysis is enabling the identification of distinct molecular subgroups of tumors with unique susceptibilities to standard treatments, also known as “genome-guided treatment” [63,64]. Additionally, AI-assisted radiomics is being used for guidance during the treatment path. For example, DL algorithms and neural networks trained on CT images of patients who achieved a complete response to chemotherapy have demonstrated superior predictive accuracy compared to traditional methods in identifying patients likely to respond to chemotherapy in colorectal cancer [63,65]. By identifying patients with differing likelihoods of complete responses ahead of time, it provides an opportunity to consider differing modalities on a case-by-case basis. These abilities, along with AI’s role in pathomics and the other applications highlighted in this review, provide a scaffold on which a new approach to precision medicine can be built.

While the application of AI has thus far been discussed in a segmented manner, it is possible that the greatest potential lies in the integration of a multiomic approach. Though individual AI applications can reduce burden and enhance accessibility through improvements in accuracy and efficiency, it is the combination of approaches that confers the greatest potential for innovation in HNC care. AI’s ability to process large volumes of data and its broad applications in HNC create the potential for algorithms capable of revision and personalization of treatment plans by using this multiomics approach. As illustrated in Figure 3, this cyclical process of data input, analysis, optimization, collection, and re-entry allows for continued refinement and personalization of precision medicine for HNC patients.

## 7. Ethical, Practical and Legal Considerations

The integration of AI into medical practice, particularly in the management of HNC, has lead to transformative advancements in diagnostics, prognostics, and personalized treatment planning [5]. However, the utilization of AI in healthcare raises serious ethical, legal, and practical concerns, all of which center around data privacy, algorithmic bias, and clinician trust that must be addressed to ensure its responsible and equitable application. This section examines these considerations with regard to four crucial areas: maintaining patient privacy in compliance with the Health Insurance Portability and Accountability Act (HIPAA), algorithmic bias, risks associated with AI processing inaccurate EHR data, and both public and professional distrust stemming from limited understanding of ML.

AI systems processing sensitive patient data must adhere to HIPAA regulations to protect privacy and maintain trust [5]. EHRs, imaging, and wearable data devices contain protected health information, necessitating reliable encryption and access controls [66]. Non-compliance risks data breaches, violating patient confidentiality and legal protections. Patients’ apprehensions to AI-driven data handling reflect the need for transparent governance frameworks [67]. Implementing and maintaining HIPAA-compliant systems requires significant resources but is critical for ensuring ethical AI utilization and EHR stewardship.

Algorithmic bias poses a significant ethical challenge, as models trained on non-representative datasets may develop algorithms that produce inaccurate output [5]. For example, if a dataset disproportionately represents certain patient demographics, comorbidities, or age groups, the AI generated algorithm may lead to disparities in diagnostic accuracy or treatment recommendations. This can reduce the generalizability of the algorithms and potentially lead to poorer clinical decisions for patient groups not represented in training datasets [68]. This type of error occurs through revealing the biases currently present in our healthcare system and amplifying them during the training process of ML models [69,70].

In a study by Koyuncu CF et al., researchers used computational image analysis of digitized histopathological slides from 136 patients (68 Black and 68 White) with HPV-associated OPSCC to quantify multinucleated tumor cells, a feature linked to tumor aggressiveness. The study used a multinucleation index (MuNI), an image analysis-derived metric that has previously been prognostic in HPV-associated OPSCC patients. The analysis revealed that Black patients had significantly higher multinucleation rates than White patients, which correlated with worse clinical outcomes, including poorer survival rates. However, the thresholds the model used to develop statistically relevant prognostic data were derived from a dataset comprising an overwhelmingly white population (95% White, 5% Black) and was only statistically prognostic for White patients despite still being able to detect differences in the MuNI for Black patients [71]. These findings suggest that multinucleation algorithms have the ability to serve as valuable prognostic tools, but only for populations adequately represented in the training and validation datasets. This study serves as an example of how AI models may propagate biases present in datasets used to train and validate algorithms if measures are not taken to carefully vet the datasets used for developing prognostic models. This highlights the need to scrutinize the data used to build these models and exercise caution when making conclusions about the generalizability of these models.

Similarly, AI models trained on EHR systems containing inaccurate patient information, charting or documentation errors, and low-quality imaging studies may exacerbate these limitations, compromising their diagnostic and predictive capabilities [70,71]. In HNC, where diverse patient populations are common, biased models can disproportionately affect certain groups if ML datasets are not carefully surveilled ahead of time for bias, inaccurate EHR data, and suboptimal imaging quality.

Mitigating bias requires diverse and inclusive training datasets along with routine model audits [72]. AI frameworks that responsibly incorporate interdisciplinary expertise are essential to ensuring the equitable application of AI models into HNC care [72]. Addressing bias is not only imperative from an ethical standpoint, but also for the enhancement of the generalizability and reliability of AI models.

Ultimately, the reliability of AI models depends on the quality of the input data. Inaccurate or incomplete EHR documentation can lead to flawed predictions and misinformed clinical decisions [5]. Errors in recording patient symptoms or treatment outcomes may skew risk assessments, reducing model reliability [70]. Standardizing data collection protocols and implementing rigorous validation processes are necessary to improve data quality, although these efforts may require substantial time and resources [73].

Distrust in AI-driven by limited understanding of ML and opaque algorithms non-open AI models-hinders its clinical adoption [5]. Clinicians may be hesitant to utilize AI tools if model outputs lack easy interpretability, while patients may be reluctant out of fear of an over-reliance on technology [74]. Explainable AI models, which provide transparent rationales for predictions, are critical to building trust amongst clinicians and patients alike [71,75]. For example, ensemble learning models, which combine multiple individual models to create more accurate and easily interpretable outputs, have improved clinician confidence in oral cancer screening [73]. Education on ML principles and user-friendly interfaces can further foster its acceptance as a useful tool for improving patient data analysis.

Explainable AI (XAI) has emerged as a critical approach to addressing the “black box” problem in modern machine learning by providing tools that make complex models more transparent. Among the most widely used techniques are Local Interpretable Model-agnostic Explanations (LIME) and SHapley Additive exPlanations (SHAP). LIME works by perturbing input data and building simple, local surrogate models to approximate the behavior of the larger AI system, allowing users to see which features most strongly influenced a specific prediction. SHAP, in contrast, applies principles from cooperative game theory to calculate each feature’s marginal contribution to the overall prediction, producing a consistent and theoretically grounded measure of feature importance. Together, these methods enhance interpretability by clarifying what factors the model considers most salient, thereby improving trust and usability in clinical contexts. However, it is important to recognize that both LIME and SHAP provide approximations of model behavior rather than a direct view into the model’s internal reasoning, highlighting both the progress and ongoing challenges in the development of XAI [6].

Overall, AI’s transformative potential in HNC prognosis lies in its ability to quickly integrate multimodal data, enabling precise and personalized care. Survival prediction models help leverage clinical and molecular insights to guide treatment planning, while recurrence risk assessments and monitoring tools enhance post-treatment surveillance. However, ethical and practical challenges must be addressed to realize these benefits fully. Data privacy concerns necessitate stringent HIPAA compliance, while algorithmic bias requires proactive measures to ensure health equity. Poor data quality and distrust in AI further complicate implementation, underscoring the need for standardized protocols and explainable models. These challenges highlight the complexities involved in integrating AI into clinical practice [76]. While AI has shown undeniable promise, addressing these concerns will be essential to ensuring its responsible and effective use in medicine.

## 8. Emerging Technology and Applications of AI

Since AI’s popularization and first widespread application with large language models such as OpenAI’s ChatGPT (GPT-4.5, USA), the field has rapidly progressed from its initial limitations in speed, complexity of answers, and narrow focus to a relatively larger host of applications. These advances are currently primarily limited by availability of high-quality data in real time, computational power (hardware, energy), and high-quality programming (algorithms). Though limited by the aforementioned constraints, applications of artificial intelligence within the field of healthcare have exploded.

Some of the existing applications for AI in oncology include novel innovations such as improvements in medical imaging analysis, liquid biopsy, symptom tracking, medical chatbots, drug design, and transcriptome analysis. Although the industry has already embraced the use of AI in functions such as early screening tests for cell free DNA in cancer detection (liquid biopsy), ongoing clinical trials are also driving the pace of development of new applications of AI in radiomics, histopathology, and molecular analysis [77,78]. Other emerging, unique applications within the field include novel algorithms designed to match patients to clinical trials, tracking symptoms using patient reported outcomes, and even monitoring nonverbal patients’ facial expressions to gauge pain level [79,80,81].

As the quest for artificial general intelligence (AGI) continues, the generalizability of AI to increasingly complex tasks may supersede the need for continued human monitoring and prompting. An integrated approach using artificial intelligence may allow for high-fidelity early cancer detection, personalized oncologic drugs tailored to a patient’s unique tumor profile, better margin detection ahead of tumor resection, and the integration of patient data into medical charts seamlessly, simply through a patient interacting with the program. It will be important along this journey to continue to exert supervision over the individual parts of the algorithm used to treat patients, as this powerful, yet promising, technology can remain opaque when source code is not examined (i.e., “black box effect”). The advent of artificial intelligence appears to beckon a new era of efficiency in medical and surgical oncology, with the hope of improving efficiency and outcomes in patients with HNSCC.

## 9. Discussion

The use of AI in medicine has gained significant interest over the past 10–15 years. As AI technology continues to evolve, it unveils new potential applications in many fields of healthcare. This evolution has led to an expansion of research regarding how AI can be applied in medicine.

### 9.1. Key Findings

Over the past 25 years, the incidence of HNC has steadily risen, resulting in an increased demand for resources and medical providers at every step of the treatment course. The growing incidence, coupled with a supply of resources and personnel that struggles to keep pace, has magnified the obstacles encountered in the care of HNC. Challenges include the insidious progression of symptoms leading to late-stage presentations, limited accessibility to diagnostic modalities such as radiology, pathology, and tumor profiling, complex treatment planning requiring continuous evaluation and optimization, delivery of the intervention itself, and vigilant post-intervention surveillance. Access to cutting-edge technologies is critical, but access to the highly skilled individuals required to implement these technologies remains an additional barrier. In malignancies like HNC, where diagnosis often occurs at later stages, time is at a premium; any obstacle encountered along this timeline may have detrimental long-term effects. Fortunately, advances in AI offer a promising means to ameliorate these challenges.

AI has demonstrated utility at each stage of the HNC treatment process. In diagnostic and non-diagnostic radiology, including CT, MRI, and PET, DL techniques have shown tremendous diagnostic accuracy, enhanced and expedited tumor segmentation, improved imaging resolution, and decreased scan acquisition times. In histopathology, the diagnostic accuracy, sensitivity, and specificity achieved by AI approaches parallel, and in some cases surpass, those seen in radiology. Non-diagnostic radiological applications, such as AI-augmented adaptive planning and automated contouring during radiation therapy, have accomplished these tasks in a fraction of the time required by traditional methods. By analyzing images at the scale of a single pixel, and using this data to efficiently uncover associations, AI holds the potential to augment daily medical practice on a large scale leading to improved quality assurance and reducing inter-operator variability.

Beyond advanced image analysis and feature extraction, the ability of AI to analyze and process large volumes of non-image-based information is equally significant. With respect to HNC, this is seen when algorithms are employed to analyze repositories of patient and tumor genomic data and the broader virtual chemical space. Such approaches have demonstrated promising potential to streamline otherwise time-intensive processes.

In addition to ML and DL techniques focused on image and data analysis, NLP represents another important AI application in HNC. Through NLP, AI can summarize and generate clinical text from EHRs, further supporting healthcare delivery.

While the AI has proven useful in single applications, its greatest potential lies in the integration of multi-omics. An algorithm which incorporates the multimodal capabilities of AI allows for an approach to personalized medicine which may confer tailored treatments previously unattainable. Through a cyclical process of continued refinement, precision medicine has the potential to approach its zenith.

### 9.2. Challenges

Although AI has demonstrated immense promise in the management of HNC, its integration into clinical practice comes with significant concerns and challenges. Issues requiring attention and evaluation include EHR data quality, algorithmic bias, transparency, information security, and financial feasibility. Particularly important is the transparency, or lack thereof, surrounding AI decision-making. The “black box effect”, in which the rationale behind AI outputs remains opaque, further fuels widespread distrust. This concept is represented below in Figure 4. While seemingly simple, answering the question “How does it work?” remains nebulous at best, and it is this opacity of AI decision-making processes that presents a major barrier to its acceptance and implementation into clinical practice.

Another critical consideration is the overall paucity of research specifically investigating AI in HNC. Among studies focused exclusively on HNC, many rely on small training datasets derived from single institutions, which raise concerns regarding generalizability and algorithmic bias. These limitations likely stem from the limited institutional access to advanced AI technologies.

The lack of accessibility to AI technology in HNC research can be attributed to several factors, including financial, data, technical, legal, and human resource limitations. Financial constraints include the high upfront costs of software and hardware, coupled with limited funding specifically for AI research. AI algorithms require high-quality data, yet HIPAA regulations and other compliance requirements can restrict the sharing of such data. Technological limitations arise when institutions lack the necessary IT infrastructure to run AI models or when integrating AI systems into existing clinical workflows and institutional software is not feasible. Regulatory constraints, such as obtaining clearance from governing bodies, may be slow or currently absent. Finally, accessibility requires specialized expertise, which is not widespread; this lack of proficiency hinders the cross-disciplinary and multi-institutional collaboration necessary for advancing AI in HNC research.

Addressing these barriers requires coordinated efforts from multiple entities. Healthcare executives, policymakers, physicians, and all members of the care team must develop a foundational understanding of AI capabilities in this field. This enhanced understanding can facilitate subsequent steps, including securing funding, optimizing technology, improving data quality, developing regulatory frameworks, and cultivating a population of AI-adept professionals, ultimately fostering a healthcare environment capable of integrating AI throughout the HNC care continuum.

## 10. Conclusions

Overall, the current body of research suggests that AI holds significant potential to lift many of the roadblocks encountered in the management of HNC. However, given the concerns regarding integration, the paucity of focused literature, and limitations in generalizability, future efforts must prioritize addressing these barriers. Expanding multi-institutional and multi-specialty collaboration, further exploring multiomics AI approaches, and ensuring transparency in algorithm development are critical steps. As access to AI technology broadens, the body of high-quality literature will continue to grow, enhancing our understanding of AI’s intricacies, realizing its maximal potential, and advancing the field of head and neck cancer toward delivering effective, efficient, personalized, and timely care.

## Figures and Tables

**Figure 1 cancers-17-03023-f001:**
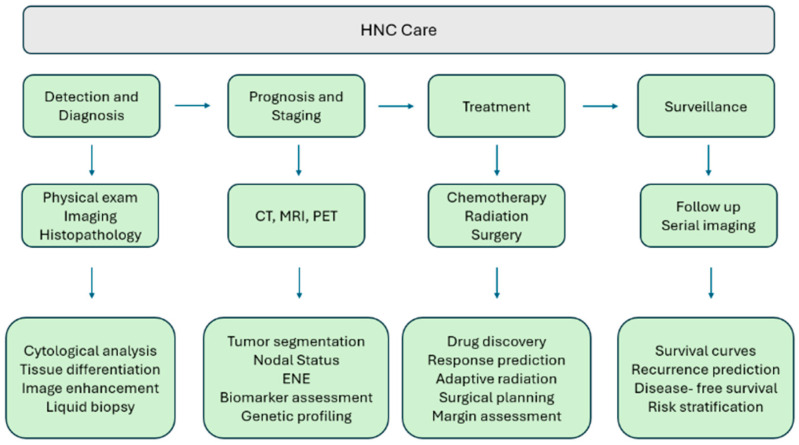
HNC roadmap and areas of AI application.

**Figure 2 cancers-17-03023-f002:**
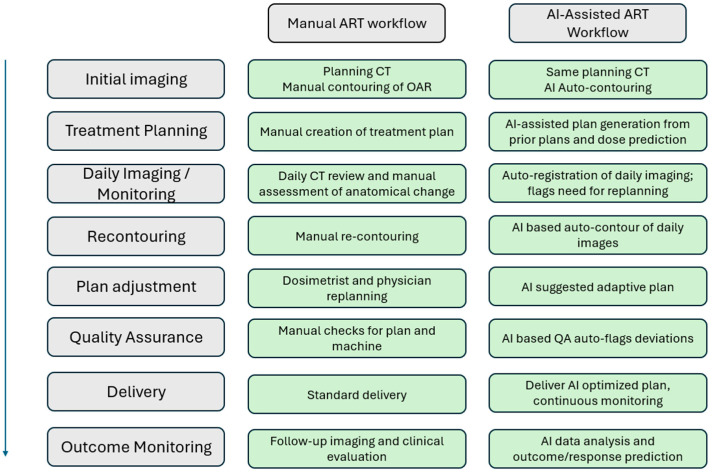
Manual versus AI-assisted adaptive radiation workflow.

**Figure 3 cancers-17-03023-f003:**
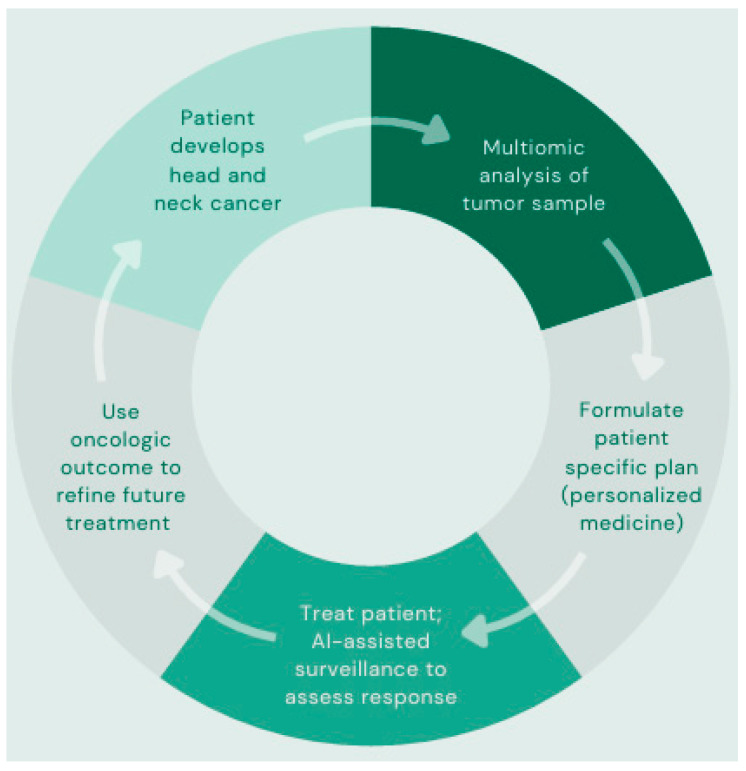
An approach to personalized medicine using multiomics.

**Figure 4 cancers-17-03023-f004:**
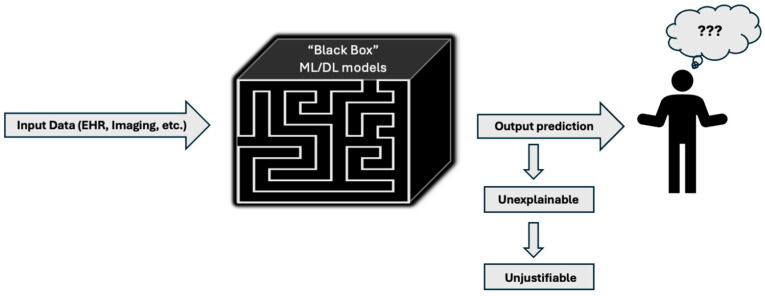
The black box effect.

**Table 1 cancers-17-03023-t001:** AI types and clinical application.

Type	Summary	Example	Reference
Supervised Learning	Trains models using labeled data, where each input is paired with a known output, to make predictions or classifications on new data	Integrated imaging and clinical data to predict PD-L1 expression using PET/CT, in NSCLC, potentially achieving AUC of 0.82–0.89 for guiding immunotherapy.	[6]
Unsupervised Learning	Discovers patterns and structures in unlabeled data without predefined outputs	Auto-encoder networks were used to extract deep features from hyperspectral images to identify tumor margins in head and neck cancer patients, achieving sensitivity of 92.32% and specificity of 91.31%.	[10]
Reinforcement Learning	Learns optimal actions through trial-and-error interactions with an environment, guided by rewards and penalties	Has been used to automate intensity-modulated radiation therapy planning by adjusting objective functions to balance target coverage and organ-at-risk sparing in radiotherapy for HNC.	[11]
Foundation Models	Large-scale, pre-trained deep learning models that can be fine-tuned for various tasks, often handling multimodal data integration for generalizable insights	In HNSCC, Foundation model-based multiple instance learning predicted 2-year overall survival from routine imaging across external cohorts with an AUC of 0.75–0.84.	[12]

**Table 2 cancers-17-03023-t002:** Commonly encountered algorithms in HNC.

Algorithms	Type	Summary	Uses in HNC
Decision Tree	Supervised ML	Uses a branching, treelike structure to make binary decisions	Tissue classification, outcome prediction
Naïve Bayes	Supervised ML	Uses probability and Bayes’ theorem to classify data	Biomarker based classification, histopathology categorization
K-Nearest Neighbor (KNN)	Supervised ML	Classifies a sample based on the majority class of its closest neighbors in the dataset	Image classification, histopathology slide analysis
Support Vector Machine (SVM)	Supervised ML	Finds boundaries (hyperplanes) that separates data into categories	Distinguishing malignant vs. benign lesions, radiomic diagnosis
Random Forest	Ensemble ML	Builds many decision trees and combines results to improve accuracy and reduce bias	Prognostication, risk stratification, treatment response prediction
Gradient Boosting Machine (GBM)	Ensemble ML	Builds many decision trees sequentially, each tree correcting the errors of the previous	Survival prediction, recurrence risk assessment, treatment planning
Artificial Neural Network (ANN)	DL	Mimics brain neurons with layers of interconnected nodes to learn complex patterns	Treatment outcome prediction, biomarker discovery, risk modeling
Deep Convolutional Neural Network (CNN/DCNN)	DL	Specialized neural network for image analysis; employs convolutional layers for feature detection	Radiology and histopathology image interpretation, tumor detection, precision diagnostics
3D U-Net	DL	Specialized CNN that captures 3D spatial features	Tumor delineation, radiotherapy planning, organ-at-risk segmentation
Generative Adversarial Network (GAN)	DL	Uses a generator and discriminator neural network to generate synthetic data or enhance images	Improving image resolution, generating synthetic pathology/radiology images

**Table 3 cancers-17-03023-t003:** Top performing algorithms.

Reference	Algorithm	Model	Performance
[14]	CNN	ResNet50	Differentiated benign vs. malignant thyroid tissue with an accuracy of 0.874
[16]	ANN	DualNet	Predicted nodal status and ENE of HNSCC with an AUC of 0.89
[27]	CNN	VGG-16	Diagnosed PTC based on cytology with an accuracy of 97.66%
[28]	CNN	VGG-19	Diagnosed and differentiated thyroid neoplasms based on cytology with an accuracy of 97.34%
[29]	ANN	AlexNet/ResNet-18 hybrid	Diagnosed OSCC based on cytology with an accuracy of 99.1%
[54]	RF	QuHbIC	Predicted outcomes of patients with p16 + OSCC based on cytology with an accuracy of 87.5%
[57]	RF	*	Predicted malignant transformation or oral leukoplakia based on cytology with an AUC of 0.84
[60]	GBM	*	Predicted 5- and 10-year recurrence rates for OSCC based on cytology with accuracies of 81.8% and 80%, respectively

‘*’ indicates an unnamed algorithm.

## Data Availability

No new data were created or analyzed in this study. Data sharing is not applicable to this article.

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
