# Peer review of "Artificial Intelligence in Head and Neck Cancer: Towards Precision Medicine"

_cancers, 2025, doi:10.3390/cancers17183023_

Round 1
Reviewer 1 Report
Comments and Suggestions for Authors
The main question of the review was to provide an overview of the application of AI in head and neck cancer care. Given that AI is rapidly emerging and integrating into clinical care, this is a timely article to both give an overview and provide specific examples that are applicable to head and neck cancer care. Specifically, this article adds examples of how AI has already been integrated into clinical care and it suggests areas where it can be further explored. The conclusions are consistent with the data provided. References are appropriate.
If there one area where I wanted the authors to elaborate more on was the topic of applying AI to surgical approaches.
Author Response
The main question of the review was to provide an overview of the application of AI in head and neck cancer care. Given that AI is rapidly emerging and integrating into clinical care, this is a timely article to both give an overview and provide specific examples that are applicable to head and neck cancer care. Specifically, this article adds examples of how AI has already been integrated into clinical care and it suggests areas where it can be further explored. The conclusions are consistent with the data provided. References are appropriate.
If there one area where I wanted the authors to elaborate more on was the topic of applying AI to surgical approaches.
- Thank you for the comment. Three additional references which expanded on the application in the surgical field have been added.
Reviewer 2 Report
Comments and Suggestions for Authors
- This narrative review is conducted to highlight current and potential applications of AI in HNC patients. This is an important topic for medical science, clinical practise and society. A narrative review is of course much more flexibel than a systematic review. Yet a design of the study / the process for identifying the literature search (such as time period of importance, type of publication, databases to be consulted) is lacking. Also the chosen subtopics seem to be organized by association.
- Section 1 discusses the epidemiology of HNC. In line 47 reference 2 is quoted providing a pseudo accuracy in the healthcare costs. The refence is from 2014 and was based on data of medical expenditure panel surveys from 2006, 2008 and 2010. So this data is limited relevant in 2025.
- Section 2 reads as really (too) simply put. That AI was first introduced in 1956 (line 90) is debatable. Ref 7 is to an internet blog source. Peer reviewed references are available, hoewever, the paper provides no plan.
- line 413 The paper states that the greatest potential of AI lies in the integration of the multiomic approach. There is no proof provided. This may be true but is not helping the HNC community for years to come. Adding much more information and thus variables in for example the treatment decision plan will it make even more difficult as the clinical interpretation is more challenging. (Similar to why segmentation on MRI is more difficult in comparison to CT, the signal expression is richer with MRI).
- line 591 Concerning AI and HNC studies paper states that many rely on small training datasets derived from single institutions, which raise concerns regarding generalizability and algorithmic bias. And the continues with "These limitations likely stem from the limited institutional access to advanced AI technologies." Why is that the cause of this apparent problem? And what measures can be identified to solve this? The answer would be a contribution of this paper!
Author Response
Thank you for your suggestions.
A study design of this narrative review was suggested
- This narrative review was conducted using literature searches of PubMed, ResearchGate, and scopus [2014-2025]. Boolean operators were used to identify relevant peer-reviewed articles, including both original studies and reviews. Keywords included in search are listed below in “keywords” section*. Reference lists of selected papers were also screened.
Section 1 discusses the epidemiology of HNC. In line 47 reference 2 is quoted providing a pseudo accuracy in the healthcare costs. The refence is from 2014 and was based on data of medical expenditure panel surveys from 2006, 2008 and 2010. So this data is limited relevant in 2025.
- This section was removed as we were unable to find more recent data
Section 2 reads as really (too) simply put. That AI was first introduced in 1956 (line 90) is debatable. Ref 7 is to an internet blog source. Peer reviewed references are available, however, the paper provides no plan. line 413 The paper states that the greatest potential of AI lies in the integration of the multiomic approach. There is no proof provided. This may be true but is not helping the HNC community for years to come. Adding much more information and thus variables in for example the treatment decision plan will it make even more difficult as the clinical interpretation is more challenging. (Similar to why segmentation on MRI is more difficult in comparison to CT, the signal expression is richer with MRI).
- Revised section 2
- The reference to the internet blog was deleted
- Reworded line 413 to reflect a suggestion rather than statement
line 591 Concerning AI and HNC studies paper states that many rely on small training datasets derived from single institutions, which raise concerns regarding generalizability and algorithmic bias. And the continues with "These limitations likely stem from the limited institutional access to advanced AI technologies." Why is that the cause of this apparent problem? And what measures can be identified to solve this? The answer would be a contribution of this paper!
- This was expanded on in the challenges section
Reviewer 3 Report
Comments and Suggestions for Authors
The review is solid and comprehensive, covering a lot of ground in AI for head and neck cancer. It's clear a lot of work went into summarizing the literature, but it could be so much more useful with some visual polish. Right now, the paper only has one figure, which feels like a missed opportunity. A great first step would be to add a big, overarching figure at the beginning that maps out the entire HNC treatment process and highlights exactly where AI fits in. Think of it as a “roadmap” for the reader.
Section 4 on adaptive radiation therapy is a perfect place for a new figure. A simple side-by-side diagram comparing a manual workflow to an AI-assisted one would really drive home the benefits of AI in terms of efficiency and precision.
The paper really needs tables. They're a fantastic way to condense information and make it easy for readers to find what they need. Right now, there's no way to quickly compare studies or algorithm performance.
Speaking of which, the discussion on AI algorithms is a bit sparse. It would be super helpful to have a dedicated table that lists the key algorithms, explains them simply, and points to their specific uses in HNC.
It would be great to see a more direct discussion on the current state-of-the-art AI algorithms and their actual performance metrics in head and neck cancer. For example, mentioning that a specific DCNN model achieved a certain accuracy for a particular task would be much more impactful.
A table summarizing the different types of AI like supervised, unspurvised, rairnforcment learning as well as touching on foundation model (refer to the example studies listed below) would be a great addition. The authors could briefly describe each one and provide an HNC example to make the technical concepts clearer for a broader audience.
Please include these two studies and see the visuals and kind of tables to enrich your paper (a) From Classical Machine Learning to Emerging Foundation Models: Review on Multimodal Data Integration for Cancer Research; b) Progress and challenges of artificial intelligence in lung cancer clinical translation.
The challenges, limitations and future directions need to be summarized in a better way. This would be a great way to leave the reader with a clear takeaway.
The "black box" nature of AI is a crucial point, but it's not well-illustrated. A figure that visually represents this concept, maybe a diagram with an input going into a mystery box and then a result coming out would be very powerful.
Finally, since the "black box" is mentioned, the paper should also introduce some of the common Explainable AI (XAI) methods like LIME and SHAP. Explaining what these methods do would show the reader that solutions to this problem are being explored and are an active area of research.
Author Response
A roadmap to HNC care and the role of AI was suggested. Please see the manuscript for the figure created.
Section 4 on adaptive radiation therapy is a perfect place for a new figure. A simple side-by-side diagram comparing a manual workflow to an AI-assisted one would really drive home the benefits of AI in terms of efficiency and precision. Please see the manuscript for the figure created.
A table on common AI algorithms employed in HNC was suggested. Please see the manuscript for this table.
A table including the state-of – the – art algorithms and their specific performance was requested. Please see the manuscript for this table.
A table summarizing the different types of AI like supervised, unsupervised, reinforced learning as well as touching on foundation model (refer to the example studies listed below) would be a great addition. Please see the manuscript for this table.
The challenges, limitations and future directions need to be summarized in a better way. This would be a great way to leave the reader with a clear takeaway.
- These topics were revised accordingly and expanded upon in the challenges section
The "black box" nature of AI is a crucial point, but it's not well-illustrated. A figure that visually represents this concept, maybe a diagram with an input going into a mystery box and then a result coming out would be very powerful.
- Figure added
Finally, since the "black box" is mentioned, the paper should also introduce some of the common Explainable AI (XAI) methods like LIME and SHAP. Explaining what these methods do would show the reader that solutions to this problem are being explored and are an active area of research.
- Added these topics and the 2 citations suggested